# Pragmatic Expectancy on Microbiota and Non-Small Cell Lung Cancer: A Narrative Review

**DOI:** 10.3390/cancers14133131

**Published:** 2022-06-26

**Authors:** Giulia Maria Stella, Filippo Scialò, Chandra Bortolotto, Francesco Agustoni, Vincenzo Sanci, Jessica Saddi, Lucio Casali, Angelo Guido Corsico, Andrea Bianco

**Affiliations:** 1Department of Internal Medicine and Medical Therapeutics, University of Pavia Medical School, 27100 Pavia, Italy; vincenzosanci01@universitadipavia.it (V.S.); corsico@univp.it (A.G.C.); 2Unit of Respiratory Diseases IRCCS Policlinico San Matteo Foundation, Department of Medical Sciences and Infective Diseases, 27100 Pavia, Italy; 3Department of Translational Medical Sciences, University of Campania “L. Vanvitelli”, 80138 Naples, Italy; filippo.scialo@unicampania.it (F.S.); andrea.bianco@unicampania.it (A.B.); 4Ceinge Biotecnologie Avanzate s.c.a.r.l., 80145 Naples, Italy; 5Department of Clinical-Surgical, Diagnostic and Pediatric Sciences, University of Pavia Medical School, 27100 Pavia, Italy; c.bortolotto@smatteo.pv.it; 6Unit of Radiology, Department of Intensive Medicine, IRCCS Policlinico San Matteo Foundation, 27100 Pavia, Italy; 7Unit of Oncology, Department of Medical Sciences and Infective Diseases, IRCCS Policlinico San Matteo Foundation, 27100 Pavia, Italy; f.agustoni@smatteo.pv.it; 8Radiation Therapy IRCCS Unit, Department of Medical Sciences and Infective Diseases, Policlinico San Matteo Foundation, 27100 Pavia, Italy; j.saddi@smatteo.pv.it; 9University of Milano-Bicocca, 20900 Monza, Italy; 10Honorary Consultant Student Support and Services, University of Pavia, 27100 Pavia, Italy; lucio.casali@unipv.it

**Keywords:** lung cancer, microbiota, personalized medicine

## Abstract

**Simple Summary:**

Growing evidence suggests that the microbiota—or better, the changes in microbiota composition and characteristics—plays a role in lung cancer onset and is associated with each phase of tumor progression. A relevant amount of data are now available and under investigation regarding the characterization of the microbiome associated with lung cancer. However, in some cases, they are redundant, and in many others, hyper specialist and technical. The goal of this review is to summarize and discuss the state of the art regarding the cross-talk between lung tumors and the microbial compartment and also to put into a clinical frame their mutual interaction and influence. This pragmatic approach will be of help for future diagnostic, therapeutic and prognostic purposes.

**Abstract:**

It is well known that lung cancer relies on a number of genes aberrantly expressed because of somatic lesions. Indeed, the lungs, based on their anatomical features, are organs at a high risk of development of extremely heterogeneous tumors due to the exposure to several environmental toxic agents. In this context, the microbiome identifies the whole assemblage of microorganisms present in the lungs, as well as in distant organs, together with their structural elements and metabolites, which actively interact with normal and transformed cells. A relevant amount of data suggest that the microbiota plays a role not only in cancer disease predisposition and risk but also in its initiation and progression, with an impact on patients’ prognosis. Here, we discuss the mechanistic insights of the complex interaction between lung cancer and microbiota as a relevant component of the microenvironment, mainly focusing on novel diagnostic and therapeutic objectives.

## 1. Introduction

Lung cancer is one of the most frequent causes of death for solid tumors. Despite the relevant progress in the personalized approach to lung cancer, patient survival is still poor. Lung cancer has different pathologic features. With respect to non-small cell lung cancer (NSCLC), over the past decades, in-depth analyses of tumor genomes and signaling pathways have further defined distinct diseases with genetic and cellular heterogeneity [1,2]. Moreover, the tumor surrounding microenvironment emerges as a key player in tumor progression, and the advent of immunotherapy has led to a complete change of the paradigm frame in which the disease is clinically managed [3,4]. The composition of the tumor microenvironment includes non-malignant cells, secreted proteins and blood vessels, as well as an extracellular matrix. Both cancer and its surrounding stroma interact with the resident microbiota, which features dynamic and plastic properties that modulate both cancer progression and host immune responses. A large amount of recent data point out significant insights in the cross-talk between cancer, microbiome and immune axis, which can be exploited to improve the personalized approach to the disease [5,6,7,8]. 

The microbiota is usually defined as the assemblage of microorganisms, namely the microbiome, living in a defined environment. The definition does not include phages, viruses, plasmids, prions, viroids and free DNA [9]. On the other hand, the term microbiome encompasses not only all living microorganisms but also their microenvironmental and surrounding biologic context, which is defined by the metabolites they produce [10]. The first and most known example of an interaction between microorganisms and cancer relates to *Helicobacter pylori (HP)* as the dominant species of the human gastric microbiome. Indeed, *HP* colonization causes a persistent inflammatory response, and *HP*-induced gastritis is the strongest singular risk factor for cancers of the stomach [11,12,13]. From initial observation, it was conceivable that the carcinogenic risk was modified by the active interaction between strain-specific bacterial components, host responses and/or specific host–microbe interactions. This first observation allowed the subsequent identification of bacterial and host mediators that augment gastric cancer risk and a focus on prevention approaches and screening programs that target *HP*-infected populations, being at a higher risk of cancer development, as well as the provision of mechanistic insights into inflammatory-induced carcinomas developed beyond the gastric niche [14,15,16,17]. 

It has been already demonstrated that the microbiome found in NSCLC patients is similar in composition to that of healthy subjects even though with a higher grade of dysbiosis due to the rare species, e.g., *Lactobacillus rossiae*, *Bacteroides pheteryogenes*, *Paenibacillus odorifer*, *Pseudomonas entomophila*, *Magnetospirillum gryphiswaldense*, *fungus Chaetomium globosum* [18]. In healthy subjects, the microbiota composition is modulated by the effects of different conditions, which include aspiration of oral secretions, the interaction with the host immune system [19] and the constant interaction between pulmonary and distal microbiota, mainly that in the gut. The metabolites that are produced and released may affect cancer onset by: (i) directly promoting cell malignant transformation; (ii) modulating the local immune microenvironment through promotion of immune–inflammatory reactions; (iii) regulating systemic immune response (Table 1) [20]. Many reports are already available regarding the NSCLC-associated microbiota (for a review, see Refs [21,22,23,24]). Based on the already available amount of data, some issues remain to be clarified based on two different hints: (i) to avoid the production of redundant data; (ii) to decipher the impact of the microbiome on tumor evolutionary trajectory [25,26] and clinical progression. 

## 2. Methods

We performed a systematic review of published data by consulting the following databases: Scopus, Web of Science, Cochrane, Google Scholar and PubMed. The last search was run on 5 March 2022. To assure the highest sensitivity, we selected the following keywords: lung cancer OR NSCLC AND microbiome OR microbiota AND immune checkpoint inhibitors AND genetics AND radiation therapy; OR NSCLC AND microbiome AND computed tomography AND radiomics. Two authors, G.M.S. and V.S., independently screened the titles of the selected publications. G.M.S. and J.S. independently read abstracts of the selected papers, and subsequently, the full text of those studies whose abstracts passed the screening. Any disagreement was discussed, and a mutual agreement was finally reached. 

## 3. Microbiota Interplay with Oncogenes Activation and Tumor Onset

### 3.1. Tracking Primary Origin of Tumor Cells: Which Role for Microbes? 

Lung cancer is defined as a disease family, featuring cell lineages of different embryonic origins. NSCLC, which represents about 80% of lung cancer diagnoses, derives from cells originating from embryo endoderm, whereas small-cell lung cancer (SCLC) cells (representing the remaining 20% of cases) derive from the neural crest layer [27,28]. The advances in the knowledge of biological, genetic and molecular features of lung tumors, together with the development of diagnostic tools, led to the identification and validation of novel actionable targets, as well as the development of more effective therapeutic platforms. These approaches have led to the development of a tangible personalized medicine approach based on the fact that the detection of a specific molecular alteration identified patients that are likely to respond to those drugs that block that target. The subsequent need for clinical trials requires constant interaction among multidisciplinary expertise and various disciplines. The main and critical issue concerns the identification/confirmation of the primary site of origin of neoplastic cells, since this can effectively guide the search for “actionable” molecular targets. In other words, the detection of a mass in the lungs imposes a determination of whether it is a primary lung tumor or a lung metastatic lesion spread from a different organ of origin [29,30]. The immunophenotype expressed by cancer cells can be of help routinely in suggesting the putative primary cases [25], whereas more advanced technologies, such as tools based on epigenetic tumor profiling [31] or gene expression analysis [32], can be applied to more enigmatic cases. 

The lung microbiota in healthy adults is continuously evolving and seems to be dominated by genera *Bacteroides* and *Firmicutes* [33] and communities such as *Megasphaera*, *Streptococcus*, *Pseudomonas*, *Fusobacterium, Sphingomonas* and *Proteobacteri*, which are phylogenetically diverse from those present in the upper airways [34]. The microbiome participates in a Darwinian competition and selection of cells of origin. Several markers can be of help in determining the tissue of origin of a mass. Among them, the thyroid transcription factor-1 (TTF-1) is implicated in embryonic differentiation and morphogenesis of both the lung and thyroid. It is overexpressed in most lung adenocarcinoma, whereas squamous cancer cells are almost negative [29]. It cooperates in modulating the expression of surfactant apoproteins (A, B, C) and Clara cell antigens. TTF-1 expression can be easily determined by immunohistochemistry (IHC), which is used to differentiate those cancers that originate in lung epithelium from secondary lung lesions coming from other organs [29]. Interestingly, the expression of TTF-1 and cytokeratin 7 (CK7), which is found in simple epithelia in several organs [29], has been positively correlated with *Enterobacteriacee* presence in a context of dysbiosis of the salivary microbiome of non-smoking female lung cancer patients. In the same cancer population, the positive strain of Napsin A—which is highly expressed in lung adenocarcinomas—is used to confirm the pulmonary origin of malignant cells and is directly associated with genera *Blastomonas* [35]. The *Enterobacteriaceae* belongs to the healthy human gut microbiome and is considered commensal [36]. Although the literature data can only allow us to conclude that certain oral microbes are significantly associated with the presence of lung adenocarcinoma in non-smokers, further investigation should be addressed to evaluate if a significant variation of *Enterobacteriaceae* species, which is related to host immunity and environmental factors, might play a role in the selection of malignant cells. It should be underlined that microbial composition in the lungs is affected by the bi-directional air movement through the upper airways and the mucociliary escalator, the larynx and the alveoli that occur during each ventilatory act [37]. This is, thus, a local microenvironment that is extremely dynamic when compared to that of the gut. However, the cross-talk between microbes from the intestinal tract and the lung is widely described, and alterations of this symbiotic equilibrium are associated with several malignant and immune/inflammatory diseases [38,39,40]. Another relevant point is that alteration of the microenvironment can affect microbiome composition. With respect to the lungs, exposure to toxic agents—among which cigarette smoke plays the main role—deserves specific attention. Smoking inflames and irritates the lungs. Moreover, it acts as a carcinogen through two different kinds of effects: (i) Direct effects, which are essentially related to: (a) formation of DNA adducts that can lead to oncogene mutations (e.g., *KRAS*-Kirsten rat sarcoma virus gene), (b) hypermethylation of promoters of numerous genes (transcriptional silence of tumor suppressor genes), (c) increased chromosomal instability. Many chemical components present in the tobacco smoke can initiate the process of oncogenesis, promote the progression of existing neoplasms and/or act as co-carcinogens. The initiating action is mainly related to the neutral fraction of smoke, which is rich in polycyclic aromatic hydrocarbons; the promoting action is mainly played by the heavy acid fraction. (ii) Indirect effects are related to the increase in the (geno) toxicity of smoke—induced by molecular alterations in the DNA repair processes and in the metabolism of toxic agents associated with exposure to smoke [41,42,43]. Smoke alters not only the lung microbiome but also that of the gut and the upper respiratory tract [44]. Indeed, higher microbial diversity has been reported in animals after exposure to smoke fog, and that variation in the metabolic products, mainly in the gut, can be associated with lung cancer progression [45,46]. Overall, smoke-related damage to the local microbiome is a consequence of two different effects: (i) induction of dysbiosis due to the generation of inflammatory mediator end immune cytokine release; (ii) modulation of biofilm, which promotes the persistence of specific taxa, thus inducing chronic infections [44]. Moreover, bacteria can generate a pro-carcinogenic milieu by promoting DNA damage, which can contribute to the generation of somatic mutations [47]. This mutagenic effect can synergize with that of tobacco smoke in promoting epithelial transformation. Moreover, damages induced by smoke allow colonization of the lower respiratory tract by pathogenic microorganisms, such as *H. influenzae M. catarrhalis*, *S. pneumoniae* and *P. aeruginosa*, thus sustaining chronic inflammatory diseases and fibrogenesis, which, most often, are associated with cancer onset [48]. 

### 3.2. Genetic Signatures on Microbiota and Effect of Microbiota on Epigenetics

Cancer is a multi-hit process requiring mutations of multiple oncogenes and inactivation/deletion of tumor suppressor genes. The genetic lesions identify actionable targets with prognostic and predictive values. Oncogenes are defined by the acquisition of somatic mutations, which results in a dominant gain of function of the corresponding protein, leading to cellular aberrant proliferation. The occurrence of a single-point mutation in one allele (heterozygosity) is enough to induce oncogene activation. Therefore, tumors become addicted to the genetic alterations that are responsible for oncogene activation, which, in turn, determine the expression of proliferative signaling [2]. Mutations affecting receptor tyrosine kinases (RTK) have been demonstrated to drive many solid cancers, including NSCLC. Among RTKs, the epidermal growth factor (EGF) receptor family is the best known. It consists of four members: EGFR (ErbB1, HER1), ErbB2 (HER2, new in rodents), ErbB3 (HER3) and ErbB4 (HER4). Somatic mutations affecting the *EGFR* gene drive more than 20% of NSCLC [49], which become addicted to these lesions [50]. Downstream mediators include (i) the *KRAS*-BRAF-MEK cascade, which is involved in promoting cell proliferation and the phosphatidylinositol 3-kinase (PIK3CA)-AKT-mTOR axis, which sustains cell survival and motility [51,52].

The fact that certain microbes are associated with cancer progression is confirmed by the observation that species in NSCLC are associated with the genetic profile of neoplastic cells. The main attention has been devoted to the correlation of *EGFR* status and lung microbiome, and experimental reports already suggested that in bronchoscopy-obtained NSCLC samples, the detection of *EGFR*-mutant cells has been correlated with the abundance of rare species, such as *Rhizopus oryzae*, *Natronolimnobius innermongolicus, Staphylococcus sciuri* [18]. The genus *Parvimonas* has been documented to be significantly enriched in *EGFR*-mutant adenocarcinomas as well [53]. Similarly, in colorectal cancer, the occurrence of somatic mutations affecting the *KRAS* oncogene has been significantly associated with the composition of intestinal flora, namely *Roseburia, Parabacteroides, Metascardovia, Staphylococcus* and *Bacillales,* thus promoting cancer onset [54]. Interestingly, some microbial products, such as the short-chain fatty acid (SCFA) butyrate, are known to impair the proliferation and renewal of stem cells in the intestinal crypts, so colonocytes in the upper crypt consume and oxidize SCFAs for their metabolism and to defend the stem fraction [55]. Those *KRAS*-mutated colon cells upregulate glycolysis and lactate production and decrease oxygen consumption [56]. The latter might allow the selection into the gut microbiota of those facultative anaerobe species that do not produce SCFAs from fibers, even though some facultative anaerobes can produce SCFA [57]. Therefore, it should be hypothesized that the *KRAS*-activated signal might lead to the avoidance of the growth-inhibitory effects of butyrate. Although no direct experimental reports evaluate the role of SCFA in the lung epithelium, the observations made on the gut should be of potential interest also in the NSCLC context, since *KRAS* genetic changes are frequently detected in those tumors aroused in smoker subjects and might be kept under consideration in treatment design and approach [58,59,60]. Smoke affects bacteria in different ways [61]. Moreover, it should be highlighted that the dysbiotic aspect could be driven by a certain background. Thus, some epidemiologic evidence suggests that cigarette smoke promotes an environment, which favors strict/facultative anaerobes [62]. Moreover, the increased *KRAS*-driven ERK and PIK3CA signaling has been associated with an enrichment of oral species (*Streptococcus* and *Veillonella*) in lung tumors [63]. Coherently, in vitro exposure of airway epithelial cancer cells (A549 line, which is a model of lung cancer) to supernatants or heat-killed bacteria (*Veillonella*, *Prevotella* and *Streptococcus*) induces upregulation of ERK and PIK3CA-associated signaling cascades [63,64]. On the other hand, it has been reported that tumors harboring a mutation in the oncosuppressor TP53 gene, including lung cancer, have a unique bacterial population and a taxonomic signature dominated by *Acidovorax* spp., which is specifically associated with both squamous-cell histology and smoking habit [65]. 

The onset and progression of cancer derive from the interaction between the genetic asset of transformed cells and dynamic epigenetic lesions [66]. Gene inactivation by hypermethylation promotion is most frequently found in cancer, and abnormal methylation of the CpG islands located in the gene promoter regions leads to transcriptional silencing [67]. Post-translational histone modifications are an additional layer of epigenetic control altered during human carcinogenesis and include several modifications, such as acetylation, methylation, phosphorylation, ubiquitylation and sumoylation [68]. The microbiome not only impacts the genetic profile of lung cancer, but it can also modulate gene transcription by acting at the epigenetic level. Although the exact mechanism behind the interaction is not fully clarified, it should be noted that metabolites generated by all the microbial compartments are implicated in the maintenance of epigenetic homeostasis [69]. In the lung cancer context, dysbiosis, which is associated with malignant transformation, releases metabolites and toxins, such as acetaldehyde, with an impact on the epigenetic asset, thus cooperating with tumor progression [70]. For instance, (i) it has been demonstrated that butyrate inhibits most HDACs [71,72]. Moreover, it has been shown that the gut-flora-mediated fermentation of dietary fiber gives rise to butyrate, which seems to act in cancer prevention with two different effects. The first is the impairment of the growth of colon cancer cells and their apoptosis promotion. Moreover, through the activation of several enzymes involved in drug metabolisms, it promotes the reduction in mutational burden and, consequently, the risk of cancer development [73]. 

## 4. How, Where and When to Analyze NSCLC-Related Microbiota

Several strategies have been developed to analyze microbial expression in healthy subjects and cancer patients [74]. The first approach encompasses sequencing techniques, among which are high-throughput 16S ribosomal RNA gene amplicon sequencing (16S rRNAseq), and whole-shotgun metagenomics are the most frequently used. The 16S rRNAseq approach allows a PCR amplification of the bacterial 16S rRNA gene before sequencing. It can identify microbes at the genus level, and it is a convenient and useful approach in routine settings [75,76]. On the other hand, whole-shotgun metagenomics is applied for taxonomic profiling (diversity and abundance), as well as functional analysis, and assures a high coverage for species-level detection [77]. The application of third-generation (long-read) sequencing technologies allows rapid, precise and comprehensive analyses of metagenome and microbiome samples [78,79]. Several bioinformatic algorithms and data mining approaches—most of which have been promoted thanks to the Human Microbiome Project—have been developed to allow a functional analysis of taxonomic profiles obtained through raw data obtained by sequencing [80,81]. Moreover, for a full comprehension of metagenomic and next-generation sequencing data, conventional and biostatistical tool analyses are required to consider the cohorts’ composition and subsequent balancing and adequate study powers [82,83]. The perspectives for the near future encompass other -omics, with the main hype around the “radiomics”. Radiomics aims at extracting from medical imaging (CT, MRI, US, XR and also PET-CT) minable information, hidden from the human naked eye but capable of being caught by data characterization algorithm. This information, which can be regarded as imaging biomarkers, can be used to refine diagnosis, select the best treatment or define the prognosis of patients starting from the imaging. No attempt so far has been reported of applying radiomics in this setting, underlining one more time that microbiome is the frontline of clinical research. Nonetheless, the multi-omics approach is advocated by several authors [84], and some preliminary attempts of matching imaging and microbiome for lung disease have already been tested [85].

Concerning NSCLC, the question of sample collection for a correct microbiome analysis is of extreme relevance. The first is related to the quantity of neoplastic tissue, which is routinely available for conventional pathology and more translational purposes, such as microbiota definition. Most lung cancer patients are also affected by chronic obstructive disease (COPD) and/or emphysema with a past or current smoking habit. Most often, they also carry other smoke-related comorbidities, including chronic heart failure and systemic vasculopathy. Overall, in the clinical context, together with the fact that, in most cases, NSCLC is diagnosed in advanced disease stage [86], the tumor samples obtained by bioptic procedures are extremely limited. In this perspective, the development of the cell block technique, which consists of processing small tissue fragments derived by fine-needle aspiration to obtain paraffin block, has significantly improved diagnostic accuracy when, as in the case of lung masses, more invasive bioptic approaches cannot be performed [87,88,89,90]. Working on small samples, such as those represented by the cell block technology, has two main implications: (i) the possibility of missing tumor histological and molecular heterogeneity is overcome by performing multiple aspirates in the mass to sample different malignant subclones; (ii) the low number of total cells sampled is highly enriched in tumor cells. These issues should be underlined when considering the cancer microbiome, since diversity and composition of the microbiota across tumoral and peritumoral microhabitats have been reported. This point is strictly related to the definition of tumor microenvironment (TME), which is a heterogeneously evolving and complex entity made of non-transformed elements. Modulation of tumor microenvironment has been associated with malignant progression. The environmental niche can vary in the tumor or the stroma, and it is modulated by several aspects. Microbiome influences TME through different actions. Among them, it is reported that alterations in the composition and the flow rate of many nutrients within TME can promote tumor-aggressive potential. In fact, a reciprocal cross-talk subsists between nutrients released by cancer cells, which are implicated in the induction of an immunosuppressive TME and those nutrients, which are produced by stromal cells (e.g., fibroblasts, macrophages, etc.), which can modulate cancer cell phenotype [91]. In a similar fashion, the oxygen level in tumors and TME can affect cancer cell aggressiveness. Indeed, tumor hypoxia can promote metastatic processes through the activation of biological and genetic programs, such as invasive growth [92]. With respect to tumor microenvironment, low oxygen levels are associated with immunosuppression. Overall hypoxia is a marker of resistance to therapy, which also relies on the mechanic variation of blood pressure and different drug concentrations and cancer recurrence, also related to the selection of a stem-like phenotype [93,94]. Intratumoral microbiome could not be observed in the peritumoral stroma or distant sites, such as the gut [95,96,97]. However, it should also be reported that the gut–lung axis allows indirect modification of lung bacterial fraction, as has been shown through fecal transplantation strategies [98]. For instance, in vitro experiments documented that specific probiotic elements, such as *Bifidobacterium*, might reduce local inflammation by decreasing the concentrations of TNF-α and lipopolysaccharides (LPS), increasing dendritic cells (DCs) [99], overall displaying an anti-cancer capacity [100]. The characterization of small-sized samples implies some critical considerations: (i) the first is that it requires advanced sequencing technologies, such as NGS, which reduce the contamination risk; (ii) contamination controls should be performed in parallel; (iii) samples must be collected according to sterile procedures. Growing evidence suggests that the microflora varies during the different tumoral stages and that microbes can influence disease progression. Lung cancer often progresses quickly. Sometimes, lung cancer arises with multiple synchronous nodules or metastases, which can disseminate starting from a small parenchymal primary mass. It is thus relevant to evaluate the microbiome variation during cancer progression according to diagnostic, prognostic and predictive perspectives. It has been also reported that a specific microbiome is detectable in early stage lung cancer (identified in a CT scan as ground-glass nodules GGN) carrying different taxa signatures when compared with that extracted from normal adjacent tissue [101]. In addition, microbial dysbiosis is associated with tumor progression, clonal evolution and spreading to distant sites [102]. If malignant cells can model the microbiome [103], it has been also reported that distinct “dysbiotic signatures” [5] can act directly on tumor cells and indirectly through the suppression of immune response. Some experimental evidence documented the associations between *Mycoplasma pn*. infection and development of metastases in lung cancer [104]. In immortalized human bronchial epithelial cells and A549 adenocarcinoma cell lines, mycoplasma influences malignant transformation by activating the expression of the bone morphogenetic protein 2 (BMP2) growth factor [105]. 

The tumor-associated microbiome can support epithelial-to-mesenchymal transition (EMT) by promoting various signaling pathways and by activating transcription factors [106,107]. In detail, infections with certain pathogens (e.g., *Streptococcus pneumoniae* [108], *H. pilorii* [109], *F. nucleatum*) induce phosphorylation of E-cadherin, which is among the best known EMT promoters and which can activate the expression of EMT-related genes. Fusobacteria species are already enriched in premalignant lesions and are abundant in those cases with a worse outcome EMT [110,111]. Overall, these observations suggest a putative role of certain microbial species in activating EMT, although no experimental evidence specifically refers to the lungs. Detection of a distinct microbiome can be of help to identify those tumor cell clones, which feature the most efficient survival strategies and feature the highest level of plasticity and renewal capacity, namely the cancer stem cells (CSCs) [112,113]. Enrichment in the stem compartment assures the potential of recurrence, dissemination and of constitutive resistance to anticancer drugs [114,115]. 

As discussed above, lung carcinomas are most often diagnosed in advanced stages. Lung cancer cells reach distant sites through lymphatic and blood vessels. Mediastinal lymph node invasion identifies the first manifestation of tumor progression. The absence of nodal invasion is crucial to directing patients to surgery. Generally, neoplastic spreading via blood vessels is set early on distant sites. 

Blood is not a sterile compartment, and a microbiome can be detected in healthy subjects as well [116]. The liquid biopsy technology can reliably detect point mutations, deletions and insertions in circulating tumor DNA (ctDNA) [117]. It can be used to follow metastatic burden and clonal evolution, as well as to monitor response to targeted therapies [118,119]. Digital PCR can specifically detect cell-free microbial DNA. Microbiota exists in both tumor tissues and the blood of cancer patients [120]. Circulating microbiome DNA is an emerging paradigm for liquid biopsy [121]. Distinct circulating bacterial DNA could distinguish prostate cancer, lung cancer and melanoma patients from healthy populations CHEN [116]. Compared to matched stool and saliva samples, the absolute concentration of circulating free DNA(cfDNA) is low but significantly above the levels detected negative controls [122,123]. Overall, these preliminary data point out a strong potential for liquid biopsy technique to obtain integrative parameters to predict patients’ outcomes and, potentially, to predict response to therapies [124]. In this perspective, it should be noted that, at least in some tumor tissues, microbial abundances are more predictive of chemotherapy responses than gene expression [125]. Although no data are available in the literature so far, this observation is of extreme interest and could open the way to further non-invasive comparative analysis of blood to easily predict and monitor drug efficacy. 

**Table 1 cancers-14-03131-t001:** Schematic rationale for microbiota evaluation in lung cancer. In detail, HOW the microbiome impacts the tumor itself and the patient, in which phases of tumor progression, namely WHEN the cross-talk between microbes and tumor is active, WHERE it is active (not only in tumor mass but also in distant secondary sites) and WHY it is relevant to know it, since it affects both responses to anticancer therapies and patients’ outcome.

HOW	WHEN	WHERE	WHY
**TUMOR**	**INITIAL PHASES**	**PRIMARY MASS**	**RESPONSE TO THERAPY**
Significant association with the selection and transformation of primary cancer cells	Implication in early stage lung cancer, in areas with heavy air pollutions	Tumor mass	Association with response to targeted therapies Predictive biomarker for immune checkpoint inhibitor response and toxicity
Promotion, selection and survival tumor cell clones and cancer stem cells compartment		Peritumoral stroma	Implication in modulation of response to radiation therapy
**PATIENT**	**LATE PHASES**	**DISTANT SITES**	**OUTCOME**
Specific association between microbial species and cancer patient gender and smoking habit	Implication in promoting tumor dissemination	• BLOOD: Free microbial circulating DNA (Liquid biopsy) • GUT: lung–gut axis • UPPER AIRWAYS	Discrimination of long-term/short-term survivors

## 5. Therapeutic Perspective and Exploitation

Microbial-derived signals modulate numerous hallmarks of cancer through diverse mechanisms [126] and sustain a biological landscape that promotes tumor progression (Figure 1). Most of these findings have been observed in human and animal models of colorectal cancer. In this context, it has been reported that chronic colitis induced by enterotoxigenic *Bacteroides Fragilis* and by *E. Coli* ultimately promotes proliferative signals and cell genome instability [127,128,129]. The commensal bacteria, such as *Enterococcus Faecalis*, are involved in initiating the events that lead to colon cancer, such as chromosomal instability and the expression of progenitor and tumor stem cell markers [130]. As already discussed, *Fusobaterium Nucleatum* induces colorectal malignant transformation by activating E-cadherin through the adhesin FadA, which promotes the attachment of *Fusobacterium* on E-cadherin-expressing cells [131]. Moreover, it acts by modulating TME [132]. As discussed above, the tumor microenvironment is defined by a number of critical features. The peritumoral neo-angiogenesis can promote tumor invasive potential by activating EMT signaling pathways [133,134]. However, the imbalance of angiogenic growth factors and regulators induces diffuse hypoxic areas; moreover, the tumor mass itself can present necrosis due to rapid growth rate and insufficient and inefficient vasculature. Bacteria can affect the necrotic and hypoxic regions of tumors, and genetically engineered bacteria can be used in tandem with other therapies for inducing vasculature balance [135,136,137]. Radiation therapy (RT) can also impact the tumor microenvironment by reshaping the microbiome. On the other hand, acting on microbiota can be a strategy to overcome RT-induced toxicity, thus improving patients’ outcomes [22]. 

Immune checkpoints identify molecules that are located on cell surface, which act by sending inhibitory stimuli to reduce immune responses. The expression of these molecules is exploited by tumors to escape immune control as expedience to progress and disseminate [138]. The best known checkpoints are the cytotoxic T-lymphocyte-associated antigen 4 (CTLA-4) and the programed death 1 (PD-1) immune checkpoint and its ligand (PD-L1). The inhibitors of immune checkpoints (ICIs) are drugs that can block checkpoint proteins from binding with their ligands, thus restoring the immune response against cancer. The microbiome is implicated in response to immune checkpoint inhibitors (ICI) [139,140]. In lung cancer mouse models, it has been possible to show that *Bifidobacterium bifidum* strains synergistically cooperate with anti PD-L1 agents and oxaliplatin in reducing tumor mass [141]. In melanoma patients, fecal microbiota transplant (FMT) can restore sensitivity to ICIs in previously anti PD-1 resistant tumors [142,143]. Notably, the response to ICI is associated with the activation of intestinal dendritic cells and circulating T cells by specific intestinal microbes [144]. Moreover, some bacteria might increase sensitivity to CTL-4 blockade based on the upregulation of Tregs [145]. Innate immunity is represented by antigen-presenting cells (APCs), dendritic cells and B lymphocytes. APC interacts with adaptive immunity and modulates the differentiation of T lymphocytes [146]. In the airway epithelium, commensal bacteria modulate APCs to activate lymphocytes Th17 and to induce Th1 differentiation [147]. The strong reciprocal interaction between the microbiome and immune responses gives rise to unique bacterial signatures, which reflected the outcome of patients treated with ICIs [148]. In metastatic melanoma, patients who respond to ICIs have high concentrations of *Faecalibacterium Prausnitzii*, which resides in the gut and which ultimately modulates the immune system via the production of short-chain fatty acids (SCFAs). It has also been shown that for NSCLC, the kidney and urothelial carcinoma response to ICI is associated with the abundance of *Akkermansia Muciniphilia* and *Enterococcus Hirae*, in quite a cancer-specific manner [149]. The gut microbiota seems to be able to activate commensal-specific memory T cells that, in turn, cross-react with tumor-associated antigens. *E. Hirae* harbors a prophage encoding an MHC class I-binding protein that induces a CD8+ T cell response and cross-reacts with cancer antigens. This evidence could explain how *E. Hirae* might impact ICI treatment [150]. In conclusion, growing evidence suggests that cancer patients who are directed to ICI treatment might be stratified based on their microbiome composition, which carries predictive and prognostic potential. The identification of specific microbial signatures should therefore represent an additional therapeutic option to improve the response to immunotherapy. In this perspective, several clinical trials are ongoing (for details, www.clinicaltrial.gov (accessed on 1 June 2022)). The observational NCT04107168 trial is aimed at evaluating the microbiome as a predictive marker in patients treated with different immune checkpoint inhibitors. Similarly, the observational NCT04711330 trial evaluates the role of the microbiome in patients carrying locally advanced lung cancer and treated with durvalumab following concurrent chemotherapy, in a clinical setting recalling that of the PACIFIC study [151]. 

The microbiome is also involved in modulating responses to ICI-induced toxicities [152] and as a marker that can be potentially exploited to improve the quality of life [153]. Differences in the gut microbiome between lung cancer patients and healthy controls have been reported to be related to response to ICI treatment, as well as to the occurrence of adverse events and drug-related toxicities (NCT03688347 trial) [154]. Similarly, the phase IV study NCT04636775 aims to determine in immunotherapy-naïve lung cancer patients, who are going to be treated with ICI, whether microbiome composition can be related either to the outcome or to toxic effects. The ongoing MicroDurva study (NCT04680377) is a phase IV trial aimed at determining if the analysis of microbiome in lung cancer patients treated with durvalumab can predict patients’ response (toxicity). The prospective NCT04954885 trial has the goal to functionally analyze microbiomes as predictive and prognostic markers for lung cancer patients receiving combinatorial therapies, according to the INSIGNA protocol (which compares pembrolizumab alone as a first-line treatment, followed by pemetrexed and carboplatin, with or without pembrolizumab after disease progression). 

## 6. Conclusions 

Microbiota and lung cancer are more and more linked in terms of the reciprocal influence that could be exploited for disease diagnosis, therapy and as a prognostic marker. A successful strategy requires the integration of several competencies and multidisciplinary management as essential key points to translate the basic and preclinical findings into a patient’s bed. Lung cancer combination of microbiome regulation with chemo/immunotherapy emerges as a powerful strategy to improve clinical responses, and although some trials are already ongoing, many questions remain unanswered. Further data are required to clarify the impact of RT and that of small molecules on the lung microbiome, as well as the development and routinely suitable approaches to sampling and characterization of the microbiome before and during cancer therapies to monitor and possibly predict drug sensitivity. 

## Figures and Tables

**Figure 1 cancers-14-03131-f001:**
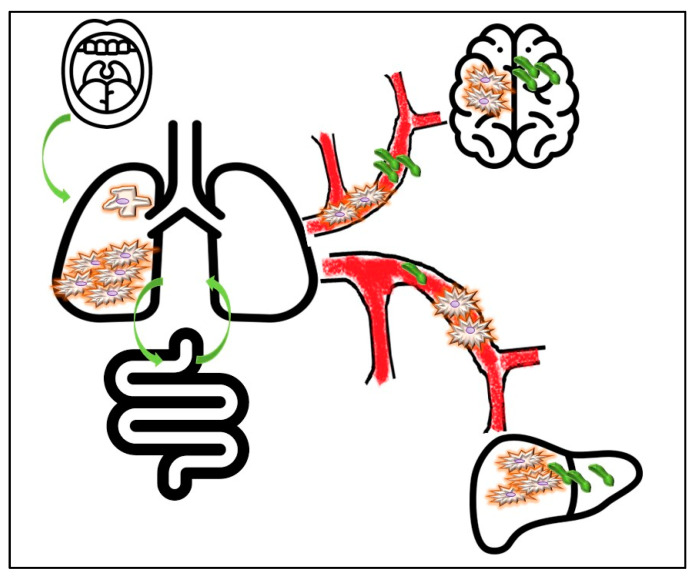
Lung cancer and microbes. Lung cancer microbiota (intratumor microbiome) composition is strictly influenced by oral cavity microbes and by the so-called gut–lung axis. Circulating and metastatic microbial signatures can be detected as well. Moreover, the microbiome also identifies an important compartment of tumor microenvironment (peritumoral and stromal microbiome), and a cross-talk exists between bacteria and cancer cells, which reciprocally influence each other with diagnostic and therapeutic implications.

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
