# Peer review of "Pragmatic Expectancy on Microbiota and Non-Small Cell Lung Cancer: A Narrative Review"

_cancers, 2022, doi:10.3390/cancers14133131_

Round 1
Reviewer 1 Report
Stella et al. present an overview of the complex relationship between microbiota and non-small cell lung cancer. Following a broad introduction of the subject, the authors delve into the role of microorganisms in carcinogenesis. Then they go over various microbiota analysis techniques that are now available, as well as their drawbacks. Finally, a summary of the microbiota's influence on therapy efficacy and ongoing clinical trials is provided. The latter point is what addd novelty to this review.
General concept comments
Apart from the opening lines, the paper requires extensive revision because it is extremely difficult to read. It lacks conceptual links necessary to explain more in details the same topic or to connect to another section. Moreover, the primary subject of each section is unclear. The authors mostly provide a list of evidences that might be retrieved by ordinary bibliographic research, and they do not build a clear scenario for the reader. In some sections statements/conclusions are too strong. References are often not correctly reported; an extensive text reorganization is recommended.
Specific comments
Line 31: It is not ‘ on the other hand’, it is another point. Modify with ‘in this context, in addition, at the same time…’
Line 56-57: is the sentence incomplete?
Ref 10: use a more updated and focused review.
Line 71-74: it is better to first refer to lung microbiota in healthy subjects (lines 75-77), and then to the dysbiotic one in NSCLC.
Table1: it is hard to understand what the purpose of this table is. Also, the layout is a bit confounding. Then, is it the rational for microbiota evaluation or evolution?
Line 83-85: which are these issues? What does ‘decipher the impact of microbiota on tumor evolutionary trajectory’ mean? I think this is obscure a deserve further clarification.
Line 120: Can the authors explain with a few more words reference 29? What is meant for advanced technologies?
Line 121-124: it is worthy avoiding strong statements. Indeed, Man et al. are also very careful: ‘lung microbiota seems to be dominated by genera…’ because it is still not certain. They also explain very well how difficult this task is, as the lung microflora is continuously evolving. In addition, I could not find most of the ‘communities’ the authors are citing in lines 122-123 in reference 30. Could the authors please add the right references? Also, do not mix genera with phyla.
Lines 124-138: the authors affirm: ‘The hypothesis to be disproven is that the composition of the lung microbiome can affect the selection of cancer cell lineage’. To this purpose, Ref. 31 is cited. In this paper, the authors affirm that is not possible to assess what comes first, ‘further studies need to be conducted to testify whether microbiome variation is a causative factor for carcinogenesis or an abductive consequence of cancer onset’. This is a very interesting report; however, the microbiome may influence cancer cell selection, but the dysbiosis could also arise because of cancer initiation. This point needs to be reported.
Line 140-142: do not use ‘can’. This relation is largely unknown; use instead a milder ‘might play’.
Line 148: Instead of commenting this data or adding other literature that would address the question, the authors jump to the effects of smoke. Please introduce it better.
Line 152-155: Please explain better point ii). Are these metabolites a consequence of exposure to smoke? I cannot find anything related to smoke and bacteria in Ref. 40.
General comments on paragraph 3.1: the authors are just adding unfocused and confusing concepts. This section should be extensively revised to clarify the different topics.
Line 160: Misleading title, as the authors are describing the effect of a genetic signature on microbiota and of microbiota on epigenetics. Please modify it.
Line 188: please remove ‘ in detail’ because this is not describing a previous assertion but rather presenting a new connected topic.
Line 191-193: This paragraph is a bit obscure. The authors first affirm that KRAS mutations upregulate glycolysis and lactate release, and reduce oxygen consumption ( Please add a reference, this is not found in the nearest ref - 50). Also, facultative anaerobes can produce SCFA and this is not at all disproven in ref 50 ( here facultative anaerobes are considered as dysbiotic markers, and in a context of Salmonella infections, a reduction in SCFA production has been observed, due to a decrease in Clostridia species). Secondly, the authors state that this could be of potential interest in NSCLC and smokers: is the lung of smokers considered aerobic or anaerobic? Can the authors add references related to this concept? Can the authors cite some papers in which the facultative aerobes or aerobic bacteria were increased in smokers’ lung? Also, smoke is really affecting bacteria in different ways ( see Nejman et al. The human tumor microbiome is composed of tumor type–specific intracellular bacteria. Science 368, 973–980. https://doi.org/10.1126/science.aay9189.). Is there a reasonable evidence of SCFAs’ effects on lung epithelium as it happens in the gut- as precisely explained in ref 49-? Please include those; otherwise, I see no benefit in discussing SCFA’s role in the gut and its oxygen levels. I would rather highlight the dysbiotic aspect that could be driven from a certain background. Also, Lines 196-197 should be moved above, in line 194, just after ref 50, but as a hypothesis, not a statement, unless properly referenced (are the species listed in ref 48 all facultative anaerobes that may contribute to a decrease in SCFA levels?).
Line 199-201. The authors are citing ref 55, which is a review, while they are talking about a specific research’s result. I guess that they are referring again to ref 54, (Tsay et al). Please correct the citations, and also the sentence which is a bit imprecise: 1) they are using A549 cells, therefore these cells are a model of lung cancer 2) Tsay and colleagues have never exposed the cells to live cells bacteria, but to supernatants or heat killed ones, which can give very different results.
Line 218: Please add a reference.
Line 221-225: please discuss better the significance of this concept or remove it. For example, butyrate is inhibiting the HDACs, thus helps in the prevention of cancer (ref 60). It is possible that some bacterial species can help in this context as butyrate producers. Has somebody proved that? The authors should go in this direction to enrich this important point.
Lined 227-242: it is a well-written and comprehensive introduction.
Lines 244-259: In general, the authors are providing a precise and specific overview of RADIOMICS. Still, they say that this technique has not been used so far by anybody, even if that would be interesting. I would then cut short this paragraph, and add this as a commentary/appendix of the previous one (related to the state-of-art techniques) describing the possibilities for the near future.
Line 252-255: Please, could the authors explain better what they mean with this sentence? What is the connection between ‘no attempts have been reported’ and ‘underlining that the microbiome is the frontline of clinical research’?
Line 257: It is not possible to get information on Akbilgic study from this reference, it seems an abstract presentation to a congress. However, it is not related to COPD. Please provide the correct reference.
Line 261-283: The authors here discuss the importance of the correct sampling, which is a crucial issue. I agree on all, I would only add few lines regarding the concept of Tumor Microenvironment. This is what shapes the microflora, therefore, what the authors state in line 277-280 is clearly obvious, as the environmental niche can vary in the tumor, or in the stroma, or, else in the intestine. Therefore, I suggest discussing better this important aspect, for example including details on nutrients, oxygen, IS effects, etc…
Line 284: from here onwards, the authors begin the description on how the microflora varies during the different tumoral stages. There is a truncation from the precedent discussion, therefore a new paragraph would help the readers in following the discussion.
Line 287: it is not ‘on the other hand’, it is something in addition to your previous statement. Also, this is a really important point, please further discuss it.
Line 296-299: this is again a strong statement. The expression ‘the tumor-promoting role of microbiota’ is way too generic. What the authors are discussing is related to one member of the Fusobacteria species (no need for italics here). Use instead ‘a putative role of certain microbial species has been discussed’. Also, the authors of reference 93, related to premalignant lesions do not refer to a healthy microflora but to infections from this bacterium. Also, as the authors are generically speaking of pathogens, they may add references regarding other species, discuss with few words and then concentrate to F. nucleatum. Apart from these issues, please add a connection with lung cancer, the subject of the review. Please, add open questions, or refer to similar evidences in lungs if there are any.
Line 317: again, discuss about the putative role of microbiota in lung cancer metastasis, or add a connection within sentences. Otherwise, the information regarding what happens in pancreatic cancer is sterile and not useful.
Line 323-329: add the reference for each topic/sentence you have written. At the end, add and discuss the perspectives this approach could open.
Line 339-342: this topic which is really interesting and deserve the addition of more details. The authors could add, for example, in which type of cancers these events have been observed.
Line 343-378: almost 40 lines with 1 reference at the end. Add proper references to each report described here. Also, please render these lines more fluid for the reader.
Line 346-349: detail a bit more the role of Bifidobacteria, or other probiotics, but keep the focus on lung cancer. For example, I would add this concept only if there are evidences about the interplay ‘probiotics in the gut - lung cancer – therapies improvement’. Otherwise, the reader would be moving far from the focus, which is related to ICI.
Line 349-378: A sequence of sentences with many different concepts. Very hard to read (considering also that there is not a reference). Make it easier by identifying the main topics of the discussion.
Author Response
Stella et al. present an overview of the complex relationship between microbiota and non-small cell lung cancer. Following a broad introduction of the subject, the authors delve into the role of microorganisms in carcinogenesis. Then they go over various microbiota analysis techniques that are now available, as well as their drawbacks. Finally, a summary of the microbiota's influence on therapy efficacy and ongoing clinical trials is provided. The latter point is what addd novelty to this review.
General concept comments
Apart from the opening lines, the paper requires extensive revision because it is extremely difficult to read. It lacks conceptual links necessary to explain more in details the same topic or to connect to another section. Moreover, the primary subject of each section is unclear. The authors mostly provide a list of evidences that might be retrieved by ordinary bibliographic research, and they do not build a clear scenario for the reader. In some sections statements/conclusions are too strong. References are often not correctly reported; an extensive text reorganization is recommended.
We thank the Reviewer for the careful revision of our work and for the fruitful comments and suggestions. The manuscript has been modified as required and is now highly enriched in scientific quality
Specific comments
Comment1. Line 31: It is not ‘on the other hand’, it is another point. Modify with ‘in this context, in addition, at the same time…’
Answer 1: we agree with this point and the text has been modified as suggested
Comment 2. Line 56-57: is the sentence incomplete?
Answer 2: we thank the Reviewer for the careful read and the sentence has been revised.
Comment 3. Ref 10: use a more updated and focused review.
Answer 3: a more recent reference has been added: Ursell LK, Metcalf JL, Parfrey LW, Knight R. Defining the human microbiome. Nutr Rev. 2012;70 Suppl 1(Suppl 1):S38-44. doi: 10.1111/j.1753-4887.2012.00493.x.
Comment 4. Line 71-74: it is better to first refer to lung microbiota in healthy subjects (lines 75-77), and then to the dysbiotic one in NSCLC.
Answer 4. We agree with this assugestion and the text has been modified accordingly, as follows: “..to that of health subject even though with a higher grade of dysbiosis due to the rare spe-cies e.g. Lactobacillus rossiae, Bacteroides pheteryogenes, Paenibacillus odorifer, Pseudomonas en-tomophila, Magnetospirillum gryphiswaldense, fungus Chaetomium globosum. In healthy subjects, the microbiota composition is modulated by the effects of different conditions, among which aspiration of oral secretions, the interaction with the host immune system , and the constant interaction between pulmonary and distal microbiota, mainly that in the gut. The metabolites that are produced and released may affect cancer onset by: i) directly promoting cell malignant transformation; ii) modulating the local immune mi-croenvironment through promotion of immune-inflammatory reactions; iii) regulating systemic immune response. Many reports are already available regarding the NSCLC-associated microbiota. Based on the already available amount of data, some issues remain to be clarified based on two different hints: i) to avoid the production of redundant data; ii) to decipher the impact of the microbiome on tumor evolutionary trajectory and clinical progression.”
Comment 5. Table1: it is hard to understand what the purpose of this table is. Also, the layout is a bit confounding. Then, is it the rational for microbiota evaluation or evolution?
Answer 5. We thank the reviewer for this comment. The table has been revised; the legend details the rationale of the table, as follows: “Table 1. Schematic rationale for microbiota evaluation in lung cancer. In detail HOW the micro-biome impacts the tumor itself and the patient, in which phases of tumor progression, namely WHEN, the cross-talk between microbes and tumor is active, WHERE it is active ( not only in tumor mass but also in distant secondary sites) and WHY it is relevant to know it: because it af-fects both responses to anticancer therapies and patients outcome”.
Comment 6. Line 83-85: which are these issues? What does ‘decipher the impact of microbiota on tumor evolutionary trajectory’ mean? I think this is obscure a deserve further clarification.
Answer 6. We thank the Reviewer for this comment. Cancer evolutionary trajectory regards the Darwinian process of genetic evolution which drive cancer progression and dissemination (Wang, S., Du, M., Zhang, J. et al. Tumor evolutionary trajectories during the acquisition of invasiveness in early stage lung adenocarcinoma. Nat Commun 11, 6083 (2020). https://doi.org/10.1038/s41467-020-19855-x; Birkbak NJ, McGranahan N. Cancer Genome Evolutionary Trajectories in Metastasis. Cancer Cell. 2020 Jan 13;37(1):8-19. doi: 10.1016/j.ccell.2019.12.004. PMID: 31935374). However we better specified the concept as follows: “Based on the already available amount of data, some issues remain to be clarified based on two different hints: i) to avoid the production of redundant data; ii) to decipher the im-pact of the microbiome on tumor evolutionary trajectory and clinical progression”
Comment 7. Line 120: Can the authors explain with a few more words reference 29? What is meant for advanced technologies?
Answer 7. We thank the Reviewer for this suggestion and the text has been modified accordingly: “The immunophenotype expressed by cancer cells can be of help routinely in suggesting the putative primary whereas more advanced technologies, such as tools based on epigenetic tumor profiling or gene expression analysis , can be applied to more enigmatic cases.”
Comment 8. Line 121-124: it is worthy avoiding strong statements. Indeed, Man et al. are also very careful: ‘lung microbiota seems to be dominated by genera…’ because it is still not certain. They also explain very well how difficult this task is, as the lung microflora is continuously evolving. In addition, I could not find most of the ‘communities’ the authors are citing in lines 122-123 in reference 30. Could the authors please add the right references? Also, do not mix genera with phyla.
Answer 8. We really thank the Reviewer for point out this criticism. The text has been revised as follows: “The lung microbiota, in healthy adults, is continuously evolving and seems to be dominated by genera: Bacteroides and Firmicutes and communities such as Megasphae-ra, Streptococcus, Pseudomonas, Fusobacterium, Sphingomonas and Proteobacteri that are phylo-genetically diverse from those present in the upper airways”.
Comment 9. Lines 124-138: the authors affirm: ‘The hypothesis to be disproven is that the composition of the lung microbiome can affect the selection of cancer cell lineage’. To this purpose, Ref. 31 is cited. In this paper, the authors affirm that is not possible to assess what comes first, ‘further studies need to be conducted to testify whether microbiome variation is a causative factor for carcinogenesis or an abductive consequence of cancer onset’. This is a very interesting report; however, the microbiome may influence cancer cell selection, but the dysbiosis could also arise because of cancer initiation. This point needs to be reported.
Answer 9. We thank the Reviewer for this suggestion and based on the other referees’ comments we modified the text as follows: “The microbiome participates in Darwinian competition and selection of cells of origin. Several markers can be of help in determining the tissue of origin of a mass. Among them, the thyroid transcription factor-1 (TTF-1) is implicated in embryonic differentiation and morphogenesis of both lung and thyroid. It is overexpressed in most lung adenocarcinoma whereas squamous cancer cells are almost negative [29]. It cooperates in modulating the expression of surfactant apoproteins (A, B, C) and Clara cell antigens. TTF-1 expression can be easily determined by immunohistochemistry (IHC), which is used to differentiate those cancers which originate from lung epithelium from secondary lung lesions coming from other organs [29]. Interestingly the expression of TTF-1 and cytokeratin 7 (CK7), which is found in simple epithelia in several organs, has been positively correlated with Enterobacteriacee presence in a context of dysbiosis of the salivary microbiome of non-smoking female lung cancer patients”.
Comment 10. Line 140-142: do not use ‘can’. This relation is largely unknown; use instead a milder ‘might play’.
Answer 10. The text has been revised as suggested.
Comment 11. Line 148: Instead of commenting this data or adding other literature that would address the question, the authors jump to the effects of smoke. Please introduce it better.
Answer 11. We thank the Reviewer for this comment and based on the other referees’ comments we modified the text as follows: “Another relevant point is that alteration of the microenvironment can affect microbiome composition. With respect to the lungs, exposure to toxic agents - among which cigarette smoke plays the main role - deserves specific attention. Smoking inflames and irritates the lungs. Moreover, it acts as a carcinogen through two different kinds of effects: i) direct. They are essentially related to: a) formation of DNA adducts that can lead to oncogene mutations ( e.g. KRAS - Kirsten rat sarcoma virus- gene), b) hypermethylation of promoters of numerous genes (transcriptional silence of tumor suppressor genes), c) increased chromo-somal instability. Many chemical components present in the tobacco smoke can initiate the process of oncogenesis, promote the progression of existing neoplasms and/or act as co-carcinogens. The initiating action is mainly related to the neutral fraction of smoke which is rich in polycyclic aromatic hydrocarbons; the promoting action is mainly played by the heavy acid fraction. ii) Indirect effects are related to the increase in the (geno) toxici-ty of smoke-induced by molecular alterations in the DNA repair processes and in the me-tabolism of toxic agents associated with exposure to smoke.
Comment 12. Line 152-155: Please explain better point ii). Are these metabolites a consequence of exposure to smoke? I cannot find anything related to smoke and bacteria in Ref. 40.
Answer 12. We agree with this comment. The text has been revised as indicated ( see answer 11 to comment 11)
Comment 13. General comments on paragraph 3.1: the authors are just adding unfocused and confusing concepts. This section should be extensively revised to clarify the different topics.
Answer 13. We thank the Reviewer for the careful reading of the text. The section has been extensively revised as suggested.
Comment 14. Line 160: Misleading title, as the authors are describing the effect of a genetic signature on microbiota and of microbiota on epigenetics. Please modify it.
Answer 14. The title as been modified as suggested: “Genetic signatures on microbiota and effect of microbiota on epigenetics”
Comment 15. Line 188: please remove ‘ in detail’ because this is not describing a previous assertion but rather presenting a new connected topic.
Answer 15. The sentence has been modified as indicated.
Comment 16. Line 191-193: This paragraph is a bit obscure. The authors first affirm that KRAS mutations upregulate glycolysis and lactate release, and reduce oxygen consumption ( Please add a reference, this is not found in the nearest ref - 50). Also, facultative anaerobes can produce SCFA and this is not at all disproven in ref 50 ( here facultative anaerobes are considered as dysbiotic markers, and in a context of Salmonella infections, a reduction in SCFA production has been observed, due to a decrease in Clostridia species). Secondly, the authors state that this could be of potential interest in NSCLC and smokers: is the lung of smokers considered aerobic or anaerobic? Can the authors add references related to this concept? Can the authors cite some papers in which the facultative aerobes or aerobic bacteria were increased in smokers’ lung? Also, smoke is really affecting bacteria in different ways ( see Nejman et al. The human tumor microbiome is composed of tumor type–specific intracellular bacteria. Science 368, 973–980. https://doi.org/10.1126/science.aay9189.). Is there a reasonable evidence of SCFAs’ effects on lung epithelium as it happens in the gut- as precisely explained in ref 49-? Please include those; otherwise, I see no benefit in discussing SCFA’s role in the gut and its oxygen levels. I would rather highlight the dysbiotic aspect that could be driven from a certain background. Also, Lines 196-197 should be moved above, in line 194, just after ref 50, but as a hypothesis, not a statement, unless properly referenced (are the species listed in ref 48 all facultative anaerobes that may contribute to a decrease in SCFA levels?).
Answer 16. We thatnk the Reviewer for point out this critical issue. The text has been extensively revised, as follows: “Similarly, in colorectal cancer, the occurrence of somatic mutations affecting the KRAS on-cogene has been significantly associated with the composition of intestinal flora, namely Roseburia, Parabacteroides, Metascardovia, Staphylococcus, and Bacillales, thus promoting can-cer onset . Interestingly, some microbial products, such as the short-chain fatty acid (SCFA) butyrate, are known to impair the proliferation and renewal of stem cells in the in-testinal crypts, so colonocytes in the upper crypt consume and oxidize SCFAs for their me-tabolism and to defend the stem fraction. Those KRAS-mutated colon cells upregulate glycolysis and lactate production and decrease oxygen consumption. The latter might allow the selection into the gut microbiota of those facultative anaerobe species that do not produce SCFAs from fibers, even though some facultative anaerobes can produce SCFA [ ]. Therefore, it should be hypothesized that the KRAS activated signal might lead to the avoidance of the growth-inhibitory effects of butyrate. Although no direct experimental re-ports evaluate the role of SCFA in the lung epithelium, the observations made on the gut should be of potential interest also in the NSCLC context since KRAS genetic changes are frequently detected in those tumors aroused in smoker subjects and might be kept under consideration in treatment design and approach]. Smoke is affecting bacteria in different ways. Moreover, it should be highlighted that the dysbiotic aspect could be driven by a certain background. Thus, some epidemiologic evidence suggests that cigarette smoke promotes an environment which favours strict/facultative anaerobes “.
Comment 17. Line 199-201. The authors are citing ref 55, which is a review, while they are talking about a specific research’s result. I guess that they are referring again to ref 54, (Tsay et al). Please correct the citations, and also the sentence which is a bit imprecise: 1) they are using A549 cells, therefore these cells are a model of lung cancer 2) Tsay and colleagues have never exposed the cells to live cells bacteria, but to supernatants or heat killed ones, which can give very different results.
Answer 17. We thank the Reviewer for this suggestion and the text has been revised accordingly: “Coherently, in vitro exposure of airway epithelial cancer cells ( A549 line which is a model of lung cancer) to supernatants or heat-killed bacteria (Veillonella, Prevotella, and Streptococcus) induces upregulation of ERK and PIK3CA-associated signalling cascades “.
Comment 18. Line 218: Please add a reference.
Answer 18. The following reference has been added: Khan, F.H., Bhat, B.A., Sheikh, B.A., Tariq, L., Padmanabhan, R., Verma, J.P., Shukla, A.C., Dowlati, A., Abbas, A. Microbiome dysbiosis and epigenetic modulations in lung cancer: From pathogenesis to therapy. Semin Cancer Biol. 2021:S1044-579X(21)00199-1. doi: 10.1016/j.semcancer.2021.07.005
Comment 19. Line 221-225: please discuss better the significance of this concept or remove it. For example, butyrate is inhibiting the HDACs, thus helps in the prevention of cancer (ref 60). It is possible that some bacterial species can help in this context as butyrate producers. Has somebody proved that? The authors should go in this direction to enrich this important point.
Answer 19. We agree with this comment and the text as been revised as follows: Although the exact mechanism behind the interaction is not fully clarified, it should be noted that metabolites generated by all the microbial compartments are implicated in the maintenance of epigenetic homeostasis. In the lung cancer context, dysbiosis which is associated with malignant transformation relases metabolites and toxins such as acetal-dehyde with an impact on the epigenetic asset thus cooperating with tumor progres-sion. For instance: i) it has been demonstrated that butyrate inhibits most HDACs. Moreover, it has been shown that the gut flora-mediated fermentation of dietary fi-ber gives rise to butyrate which seems to act in cancer prevention with two different effects. The first is the impairment of the growth of colon cancer cells and their apoptosis promo-tion. Besides, through the activation of several enzymes involved in drug metabolisms, promotes the reduction of mutational burden and consequently the risk of cancer devel-opment.
Comment 20. Lined 227-242: it is a well-written and comprehensive introduction.
Answer 20. We thank the Reviewer for this comment.
Comment 21. Lines 244-259: In general, the authors are providing a precise and specific overview of RADIOMICS. Still, they say that this technique has not been used so far by anybody, even if that would be interesting. I would then cut short this paragraph, and add this as a commentary/appendix of the previous one (related to the state-of-art techniques) describing the possibilities for the near future.
Answer 21. We thank the Reviewer for pointing out this critical observation. Based on the other Referees’ suggestion we changed and reduced the section: “Perspectives for the near future encompass other -omics with the main hype around the “radiomics”. Radiomics aims at extracting from medical imaging (CT, MRI, US, XR and also PET-CT) minable information, hidden from the human naked eye but capable of be-ing catched by data-characterization algorithm. This information, which can be regarded as imaging biomarkers, can be used to refine diagnosis, select the best treatment or define the prognosis of patients starting from the imaging. No attempt so far has been reported of applying radiomics in this setting, underlining one more time that microbiome is the frontline of clinical research”.
Comment 22. Line 252-255: Please, could the authors explain better what they mean with this sentence? What is the connection between ‘no attempts have been reported’ and ‘underlining that the microbiome is the frontline of clinical research’?
Answer 22. We thanks for this suggestion and modified the text accordingly (see answer 21 to comment 21)
Comment 23. Line 257: It is not possible to get information on Akbilgic study from this reference, it seems an abstract presentation to a congress. However, it is not related to COPD. Please provide the correct reference.
Answer 23. We agree with comment and comment 22. The sentence has been removed.
Comment 24. Line 261-283: The authors here discuss the importance of the correct sampling, which is a crucial issue. I agree on all, I would only add few lines regarding the concept of Tumor Microenvironment. This is what shapes the microflora, therefore, what the authors state in line 277-280 is clearly obvious, as the environmental niche can vary in the tumor, or in the stroma, or, else in the intestine. Therefore, I suggest discussing better this important aspect, for example including details on nutrients, oxygen, IS effects, etc…
Answer 24. We thank the Reviewer for this fruitful suggestion. The text has been modified as required: “This point is strictly related to the definition of tumor microenvironment (TME), which is a heterogeneous evolving and complex entity made of non-transformed elements. Modula-tion of tumor microenvironment has been associated with malignant progression. The en-vironmental niche can vary in the tumor, or the stroma and it is modulated by several as-pects. Microbiome influences TME through different actions. Among them, it is reported that alterations in composition and the flow rate of many nutrients within TME can pro-mote tumor aggressive potential. Actually, a reciprocal cross-talk subsists between nutri-ents released by cancer cells, which are implicated in the induction of an immunosup-pressive TME, and those nutrients which are produced by stromal cells (e.g. fibroblasts, macrophages, ecc) which can modulate cancer cell phenotype . In a similar fashion, the oxygen level in tumors and TME can affect cancer cell aggressiveness. Indeed tumor hypoxia can promote metastatic processes through the activation of biological and genetic programmes as invasive growth. With respect to tumor microenvironment, low oxygen levels are associated with immunosuppression. Overall hypoxia is a marker of resistance to therapy, which relies also on the mechanic variation of blood pressure and different drug concentrations and cancer recurrence, also related to the selection of a stem-like phe-notype”.
Comment 25. Line 284: from here onwards, the authors begin the description on how the microflora varies during the different tumoral stages. There is a truncation from the precedent discussion, therefore a new paragraph would help the readers in following the discussion.
Answer 25. We thank the reviewer for this point. We changed the text as follows: “Growing evidence suggest that the microflora varies during the different tumoral stages and that microbes can influence disease progression. Lung cancer often progresses quickly. Sometimes lung cancer arises with multiple synchronous nodules or metastases can disseminate starting from a small parenchymal primary mass. It is thus relevant to evaluate the microbiome variation during cancer progression according to diagnostic, prognostic, and predictive perspectives”.
Comment 27. Line 287: it is not ‘on the other hand’, it is something in addition to your previous statement. Also, this is a really important point, please further discuss it.
Answer 27. The text has been modified as required.
Comment 28. Line 296-299: this is again a strong statement. The expression ‘the tumor-promoting role of microbiota’ is way too generic. What the authors are discussing is related to one member of the Fusobacteria species (no need for italics here). Use instead ‘a putative role of certain microbial species has been discussed’. Also, the authors of reference 93, related to premalignant lesions do not refer to a healthy microflora but to infections from this bacterium. Also, as the authors are generically speaking of pathogens, they may add references regarding other species, discuss with few words and then concentrate to F. nucleatum. Apart from these issues, please add a connection with lung cancer, the subject of the review. Please, add open questions, or refer to similar evidences in lungs if there are any.
Answer 28. We agree with this suggestion and the text has been revised as follows: “The tumor-associated microbiome can support epithelial-to-mesenchymal transition (EMT) by promoting various signaling pathways and by activating transcription factors. In detail, infections by certain pathogens (e.g. Streptococcus pneumoniae, H. pilorii , F. nucleatum) induces phosphorylation of E-cadherin, which is among the best known EMT promoters, and which can activate the expression of EMT-related genes. Fusobacteria species are already enriched in premalignant lesions and are abundant in those cases with a worse outcome EMT . Overall, these observations suggest a putative role of certain microbial species in activating EMT, although no experimental ev-idence specifically refer to the lungs. Detection of a distinct microbiome can be of help to identify those tumor cell clones which feature the most efficient survival strategies and feature the highest level of plasticity and renewal capacity, namely the cancer stem cells (CSCs) . Enrichment in the stem compartment assures the potential of recurrence, dissemination, and of constitutive resistance to anticancer drugs “.
Comment 29. Line 317: again, discuss about the putative role of microbiota in lung cancer metastasis, or add a connection within sentences. Otherwise, the information regarding what happens in pancreatic cancer is sterile and not useful.
Answer 29. We thank the Reviewer for this point and implemented the text as follows: As discussed above, lung carcinomas are - most often – diagnosed in advanced stag-es. Lung cancer cells reach distant sites through lymphatic and blood vessels. Mediastinal lymph node invasion identifies the first manifestation of tumor progression. The absence of nodal invasion is crucial to addressing patients to surgery. Generally, neoplastic spreading via blood vessels is set early on distant sites.
Comment 30. Line 323-329: add the reference for each topic/sentence you have written. At the end, add and discuss the perspectives this approach could open.
Answer 30. We thank the reviewer for this comment. References as been added and the text implemented as required: “Overall, these preliminary data point out a strong potential for liquid biopsy technique to obtain integrative parameters to predict patients’ outcomes and, potentially, to predict response to therapies . In this perspective, it should be noted that, at least in some tumor tissues, microbial abundances are more predictive of chemotherapy respons-es than gene expression. Although no data are available in the literature so far, this observation is of extreme interest and could open the way to further non-invasive compar-ative analysis of blood to easily predict and monitor drug efficacy”.
Comment 31. Line 339-342: this topic which is really interesting and deserve the addition of more details. The authors could add, for example, in which type of cancers these events have been observed.
Answer 31. The text has been modified as follows: “In this context, it has been reported that chronic colitis induced by enterotoxigenic Bac-teroides Fragilis and by E.Coli ultimately promotes proliferative signals and cell genome instability . The commensal bacteria such as Enterococcus Faecalis are in-volved in initiating the events that lead to colon cancer as chromosomal instability and the expression of progenitor and tumor stem cell markers. As already discussed Fusobaterium Nucleatum induces colorectal malignant transformation by activating E-cadhering through the adhesin FadA which promotes the attachment of Fusobacterium on E-cadherin-expressing cells . Moreover, it acts by modulating TME”.
Comment 32. Line 343-378: almost 40 lines with 1 reference at the end. Add proper references to each report described here. Also, please render these lines more fluid for the reader.
Answer 32. We thank the Reviewer for this point. Reefrences have been added.
Comment 33. Line 346-349: detail a bit more the role of Bifidobacteria, or other probiotics, but keep the focus on lung cancer. For example, I would add this concept only if there are evidences about the interplay ‘probiotics in the gut - lung cancer – therapies improvement’. Otherwise, the reader would be moving far from the focus, which is related to ICI.
Answer 33. We thank the Reviewer for this comment. The text has been extensively revised as follows: Immune checkpoints identify molecules that are located on cell surface which act by sending inhibitory stimuli to reduce immune responses. The expression of these mole-cules is exploited by tumors to escape immune control as expedience to progress and dis-seminate. The most know checkpoints are the cytotoxic T-lymphocyte-associated an-tigen 4 (CTLA-4) and the programmed death 1 (PD-1) immune checkpoint and its ligand (PD-L1). The inhibitors of immune checkpoints (ICIs) are drugs that can block checkpoint proteins from binding with their ligands thus restoring immune response against cancer. The microbiome is implicated in response to immune checkpoint inhibitors (ICI) . In lung cancer mouse models it has been possible to show that Bifidobacterium bifidum strains synergistically cooperate with anti PD-L1 agents and oxaliplatin in reduc-ing tumor mass-. In melanoma patients, fecal microbiota transplant (FMT) can re-store sensitivity to ICIs in previously anti PD-1 resistant tumors. Notably, re-sponse to ICI is associated with the activation of intestinal dendritic cells and circulating T cells by specific intestinal microbes.. Moreover, some bacteria might increase sensitiv-ity to CTL-4 blockade based on the upregulation of Tregs. Innate immunity is repre-sented by antigen-presenting cells (APCs), dendritic cells, and B lymphocytes. APC inter-acts with adaptive immunity and modulates the differentiation of T lymphocytes . In the airway epithelium, commensal bacteria modulate APCs to activate lymphocytes Th17 and to induce Th1 differentiation. The strong reciprocal interaction between the mi-crobiome and immune responses gives rise to unique bacterial signatures which reflected the outcome of patients treated with ICIs. In metastatic melanoma patients which re-spond to ICIs have high concentrations of Faecalibacterium Prausnitzii which resides in the gut and which ultimately modulates the immune system via the production of short-chain fatty acids (SCFAs). It has been also shown that for NSCLC, kidney and urothelial carcinoma response to ICI is associated with the abundance of Akkermansia Muciniphilia and Enterococcus Hirae, according to a quite cancer-specific manner. The gut microbi-ota seems to be able to activate commensal-specific memory T cells that in turn, cross-react with tumor-associated antigens. E. Hirae harbors a prophage encoding an MHC class I-binding protein that induces a CD8+ T cell response and cross-reacts with cancer anti-gens. This evidence could explain how E. Hirae might impact on ICI treatment”.
Comment 34. Line 349-378: A sequence of sentences with many different concepts. Very hard to read (considering also that there is not a reference). Make it easier by identifying the main topics of the discussion.
Answer 34. We thank the Reviewer for this comment. The text has been revised accordingly, as follows ( see also answer 33 to comment 33): In conclusion, growing evidence suggests that cancer patients who are addressed to ICI treatment might be stratified based on their microbiome composition, which carries pre-dictive and prognostic potential. The identification of specific microbial signatures should therefore represent an additional therapeutic option to improve response to immunother-apy. In this perspective, several clinical trials are ongoing ( for details www.clinicaltrial.gov ). The observational NCT04107168 trial is aimed at evaluating the microbiome as a predictive marker in patients treated with different immune checkpoint inhibitors. Similarly, the observational NCT04711330 trial evaluates the role of the micro-biome in patients carrying locally advanced lung cancer and treated with durvalumab fol-lowing concurrent chemotherapy, in a clinical setting recalling that of the PACIFIC study .The microbiome is also involved in modulating responses to ICI-induced toxicities and as a marker that can be potentially exploited to improve quality of life . Dif-ferences in the gut microbiome between lung cancer patients and healthy controls have been reported to be related to response to ICI treatment as well as to the occurrence of ad-verse events and drug-related toxicities ( NCT03688347 trial). Similarly the phase IV study NCT04636775 aims to determine in immunotherapy naïve lung cancer patients who are going to be treated with ICI if microbiome composition can be related either to the outcome and to toxic effects. The ongoing MicroDurva study ( NCT04680377) is a phase IV trial aimed at determining if the analysis of microbiome in lung cancer patients treated with durvalumab can predict patients’ response (toxicity). The prospective NCT04954885 trial has the goal to functionally analyse microbiome as predictive and prognostic mark-ers for lung cancer patients receiving combinatorial therapies, according to the INSIGNA protocol (which compares pembrolizumab alone as a first-line treatment, followed by pemetrexed and carboplatin with or without pembrolizumab after disease progression).
Reviewer 2 Report
The authors of the manuscript have appropriately reviewed the potential and, in many cases, experimentally proven interaction between lung cancer and microbiota. This review provides a comprehensive overview of the role of microbiota regarding NSCLC cancer disease. The description of the interaction between mictobiota and NSCLC cited in the text is clearly considered and widely supported by incorporation of the most representative references strengthening the scientific value and content of the written text.
However, a number of minor comments would be considered by the authors of the manuscript, such as:
1. L23-27, Simple summary, need to be substituted by the summary of your own work.
2. L57, correct the sentence “The definition does not include phages, 56 viruses, plasmids, prions, viroids, and free DNA are [55]. [ 9 ].”
3. I suggest authors afford one or two pictures to clearly describe the correlation between the microbes and signal pathway involving in NSCLC progression.
Author Response
The authors of the manuscript have appropriately reviewed the potential and, in many cases, experimentally proven interaction between lung cancer and microbiota. This review provides a comprehensive overview of the role of microbiota regarding NSCLC cancer disease. The description of the interaction between mictobiota and NSCLC cited in the text is clearly considered and widely supported by incorporation of the most representative references strengthening the scientific value and content of the written text.
We thank the Reviewer for the positive comments and for the fruitful suggestion regarding our work.
However, a number of minor comments would be considered by the authors of the manuscript, such as:
Comment 1. L23-27, Simple summary, need to be substituted by the summary of your own work.
Answer 1. Simple summary has been added as required. “Simple Summary: growing evidence suggests that the microbiota - or better the changes in mi-crobiota composition and characteristics - plays a role in lung cancer onset and is associated with each phase of tumor progression. A relevant amount of data are now available and under inves-tigation regarding the characterization of the microbiome associated with lung cancer. Howev-er, in some cases they are redundant and in many others hyper specialist and technical. The goal of this review is to summarize and discuss the state-of-the-art regarding the cross-talk between lung tumors and the microbial compartment and also to put into a clinical frame their mutual interaction and influence. This pragmatic approach will be of help for next future diagnostic, therapeutic and prognostic purposes”
Comment 2. L57, correct the sentence “The definition does not include phages, 56 viruses, plasmids, prions, viroids, and free DNA are [55]. [ 9 ].”
Answer 2. We thank the Reviewer for this point and revised the sentence
Comment 3. I suggest authors afford one or two pictures to clearly describe the correlation between the microbes and signal pathway involving in NSCLC progression.
Answer 3. We thank the Reviewer for point out this criticism. Figure 1 has been added. Figure 1. Lung cancer and microbes. Lung cancer microbiota (intratumor microbiome) composition is strictly influenced by oral cavity microbes and by the so called gut-lung axis. Circulating and metastatic microbial signatures can be detected as well. Moreover, the microbi-ome also identifies an important compartment of tumor microenvironment (peritumoral and stromal microbiome) and a cross-talk exists between bacteria and cancer cells which reciprocally influence each others with diagnostic and therapeutic implications.
Reviewer 3 Report
This review is on a very interesting topic with relevant diagnostic and therapeutic implications in lung cancer. Although the authors did a good job reviewing all the relevant literature, there are several issues with the presentation of the material that need to be addressed before the review can be suitable for publication. Following are the major issues found.
The abstract is written in a very confusing way and it should be re-worded to convey the message more clearly.
The scope and message of Table 1 “Schematic rationale for microbiota evaolation in lung cancer” is not clear at all.
Lines 56-57: the sentence is truncated.
Line 103: “different embryo origins”. Do the authors mean “cell lineages of different embryonic origin”? “Embryo origins” is not correct.
In section 3.1, it is not very clear why the authors need to write about diagnostic issues between primary lung cancers and metastases from a different organ of origin. Also, the authors state that “The hypothesis to be disproven is that the composition of the lung microbiome can affect the selection of cancer cell lineage”. This hypothesis presented by the authors is based on the findings of one study (reference 31) where a comparison of oral microbiota with cancer status showed higher incidence of certain microbes in non-smoker patients with lung adenocarcinoma. In the reference, TTF-1, CK-7 and Napsin A were used for confirmation of diagnosis of adenocarcinoma, and there is no indication that the oral microbiome has any role in the selection of cell lineage or in the “Darwinian selection of malignant cells”. This hypothesis is very speculative and not supported by any evidence, so it should not be included in the review. The only conclusion that can be drawn from the cited literature is that certain oral microbes are significantly associated with the presence of lung adenocarcinoma in non-smokers. In itself, this is a very interesting observation and worth notice without further speculations on Darwinian competition and selection of cell of origin.
In lines 164-165, the sentence “The occurrence of a single point mutation in one allele (heterozygosity) is enough to induce malignant transformation” is not correct, because a point mutation in a single oncogene is not enough to induce malignant transformation. It has been known for decades that cancer is a multi-hit process requiring mutations of multiple oncogenes and inactivation/deletion of tumor suppressor genes.
In paragraph 4, the discussion on radiomics seems to be out of the scope of the review. The authors cite three papers (reference 76, 77, and 78). Of these, reference 76 is a review advocating for integration of multi-omics. Reference 77 is about differences in microbiome in patients with or without CT-confirmed COPD, and no radiomics is mentioned. Reference 78 is about correlation of microbiome changes with the risk of cardiovascular disease, also with no radiomics mentioned. It is not clear how these studies on COPD and CVD are related to the lung microbiome and radiomics in lung cancer. If the authors want to include the mention of radiomics, they should explain better how it would be integrated in the evaluation of microbiome in the diagnosis/management of lung cancer. However, since this is a review article, I would strongly suggest to remove this section on radiomics, because there is no supporting literature cited.
Lines 284-286: the authors state that “a specific microbiome is detectable in early-stage lung cancer (identified at CT scan as ground-glass nodules GGN) carrying different taxa signatures when compared with that extracted from non-transformed areas featuring the same GGN appearance”. However, in the cited reference (ref. 87) the microbiome in malignant GGNs was compared with normal adjacent tissue, not with benign GGNs. The authors should rephrase accordingly.
Lines 310-317: the authors introduce a discussion on the role of bacteria in the metastatic spread. How this fits in the discussion of “how, where, and when to analyse NSCLC-related microbiota” is not clear. Do the authors advocate for microbiota measurements in metastatic lesions? The only reference cited in this paragraph (ref. 100) correlates bacteria species in the primary tumor with lymphatic and metastatic spread. Also, this study is on pancreatic cancer. How this paragraph fits in the section should be explained better. Alternatively, the paragraph could be moved to another section on the role of microbiome on metastatic spread.
The paragraph between lines 343-378 contains a large amount of relevant and interesting information, but it contains no references except for one (ref. 115) at the end of the paragraph. Each sentence in the paragraph should be supported by relevant references.
Author Response
This review is on a very interesting topic with relevant diagnostic and therapeutic implications in lung cancer. Although the authors did a good job reviewing all the relevant literature, there are several issues with the presentation of the material that need to be addressed before the review can be suitable for publication. Following are the major issues found.
We thank the Reviewer for the careful revision of our work which in now enriched in scientific relevance and scientific message.
Comment 1. The abstract is written in a very confusing way and it should be re-worded to convey the message more clearly.
Answer 1. We thakn the Reviewer for this comment and the abstract has been revised as follows: “Abstract: It is well known that lung cancer relies on a number of genes aberrantly expressed, be-cause of somatic lesions. Indeed the lungs, based on their anatomical features, are organs at high risk for the development of extremely heterogeneous tumors due to the exposure to several en-vironmental toxic agents. In this context, the microbiome identifies the whole assemblage of mi-croorganisms present in the lungs, as well as in distant organs, together with their structural el-ements and metabolites which actively interact with normal and transformed cells. A relevant amount of data suggests that the microbiota plays a role not only in cancer disease predisposi-tion and risk but also in its initiation and progression with an impact on patients prognosis. Here we discuss the mechanistic insights of the complex interaction between lung cancer and microbiota as a relevant component of the microenvironment, mainly focusing on novel diag-nostic and therapeutic objectives”.
Comment 2. The scope and message of Table 1 “Schematic rationale for microbiota evaolation in lung cancer” is not clear at all.
Answer 2. We agree with this point. The table and the legend have been revised, based also on Rev.1 ‘s comment. “Table 1. Schematic rationale for microbiota evaluation in lung cancer. In detail HOW the micro-biome impacts the tumor itself and the patient, in which phases of tumor progression, namely WHEN, the cross-talk between microbes and tumor is active, WHERE it is active ( not only in tumor mass but also in distant secondary sites) and WHY it is relevant to know it: because it af-fects both responses to anticancer therapies and patients outcome”.
Comment 3. Lines 56-57: the sentence is truncated.
Answer 3. The sentence has been revised.
Comment 4. Line 103: “different embryo origins”. Do the authors mean “cell lineages of different embryonic origin”? “Embryo origins” is not correct.
Answer 4. We thank the Reviewer for careful revision. The sentence has been revised.
Comment 5. In section 3.1, it is not very clear why the authors need to write about diagnostic issues between primary lung cancers and metastases from a different organ of origin. Also, the authors state that “The hypothesis to be disproven is that the composition of the lung microbiome can affect the selection of cancer cell lineage”. This hypothesis presented by the authors is based on the findings of one study (reference 31) where a comparison of oral microbiota with cancer status showed higher incidence of certain microbes in non-smoker patients with lung adenocarcinoma. In the reference, TTF-1, CK-7 and Napsin A were used for confirmation of diagnosis of adenocarcinoma, and there is no indication that the oral microbiome has any role in the selection of cell lineage or in the “Darwinian selection of malignant cells”. This hypothesis is very speculative and not supported by any evidence, so it should not be included in the review. The only conclusion that can be drawn from the cited literature is that certain oral microbes are significantly associated with the presence of lung adenocarcinoma in non-smokers. In itself, this is a very interesting observation and worth notice without further speculations on Darwinian competition and selection of cell of origin.
Answer 5. We thank the Reviewer for point out this critical issue. The section has been modified as follows ( also based on Rev1’s suggestion): Several markers can be of help in determining the tissue of origin of a mass. Among them, the thyroid transcription factor-1 (TTF-1) is implicated in embryonic differentiation and morphogenesis of both lung and thyroid. It is overexpressed in most lung adenocarcino-ma whereas squamous cancer cells are almost negative [29]. It cooperates in modulating the expression of surfactant apoproteins (A, B, C) and Clara cell antigens. TTF-1 expres-sion can be easily determined by immunohistochemistry (IHC), which is used to differen-tiate those cancers which originate from lung epithelium from secondary lung lesions coming from other organs . Interestingly the expression of TTF-1 and cytokeratin 7 (CK7), which is found in simple epithelia in several organs, has been positively corre-lated with Enterobacteriacee presence in a context of dysbiosis of the salivary microbiome of non-smoking female lung cancer patients. In the same cancer population, the positive stain of Napsin A - which is highly expressed in lung adenocarcinomas, is used to con-firm the pulmonary origin of malignant cells and is directly associated with genera Blas-tomonas .
Comment 6. In lines 164-165, the sentence “The occurrence of a single point mutation in one allele (heterozygosity) is enough to induce malignant transformation” is not correct, because a point mutation in a single oncogene is not enough to induce malignant transformation. It has been known for decades that cancer is a multi-hit process requiring mutations of multiple oncogenes and inactivation/deletion of tumor suppressor genes.
Answer 6. We thank the Reviewer for this comment. Since the paragraph was not fully adherent to the scope of the review, we removed it.
Comment 7. In paragraph 4, the discussion on radiomics seems to be out of the scope of the review. The authors cite three papers (reference 76, 77, and 78). Of these, reference 76 is a review advocating for integration of multi-omics. Reference 77 is about differences in microbiome in patients with or without CT-confirmed COPD, and no radiomics is mentioned. Reference 78 is about correlation of microbiome changes with the risk of cardiovascular disease, also with no radiomics mentioned. It is not clear how these studies on COPD and CVD are related to the lung microbiome and radiomics in lung cancer. If the authors want to include the mention of radiomics, they should explain better how it would be integrated in the evaluation of microbiome in the diagnosis/management of lung cancer. However, since this is a review article, I would strongly suggest to remove this section on radiomics, because there is no supporting literature cited.
Answer 7. We agree with the Reviwer and, based also on Rev.1 ‘s comment, we modified and shortened the text: “Perspectives for the near future encompass other -omics with the main hype around the “radiomics”. Radiomics aims at extracting from medical imaging (CT, MRI, US, XR and also PET-CT) minable information, hidden from the human naked eye but capable of be-ing catched by data-characterization algorithm. This information, which can be regarded as imaging biomarkers, can be used to refine diagnosis, select the best treatment or define the prognosis of patients starting from the imaging. No attempt so far has been reported of applying radiomics in this setting, underlining one more time that microbiome is the frontline of clinical research”.
Comment 8. Lines 284-286: the authors state that “a specific microbiome is detectable in early-stage lung cancer (identified at CT scan as ground-glass nodules GGN) carrying different taxa signatures when compared with that extracted from non-transformed areas featuring the same GGN appearance”. However, in the cited reference (ref. 87) the microbiome in malignant GGNs was compared with normal adjacent tissue, not with benign GGNs. The authors should rephrase accordingly.
Answer 8. We thank the Reviewer for raising this issue and modified the text accordingly: “Growing evidence suggest that the microflora varies during the different tumoral stages and that microbes can influence disease progression. Lung cancer often progresses quick-ly. Sometimes lung cancer arises with multiple synchronous nodules or metastases can disseminate starting from a small parenchymal primary mass. It is thus relevant to evalu-ate the microbiome variation during cancer progression according to diagnostic, prognos-tic, and predictive perspectives. It has been also reported that a specific microbiome is de-tectable in early-stage lung cancer (identified at CT scan as ground-glass nodules GGN) carrying different taxa signatures when compared with that extracted from normal adja-cent tissue “.
Comment 9. Lines 310-317: the authors introduce a discussion on the role of bacteria in the metastatic spread. How this fits in the discussion of “how, where, and when to analyse NSCLC-related microbiota” is not clear. Do the authors advocate for microbiota measurements in metastatic lesions? The only reference cited in this paragraph (ref. 100) correlates bacteria species in the primary tumor with lymphatic and metastatic spread. Also, this study is on pancreatic cancer. How this paragraph f/as its in the section should be explained better. Alternatively, the paragraph could be moved to another section on the role of microbiome on metastatic spread.
Answer 9. We thank the reviewer for this observation. The text has been extensively modified, based also on Rev.1’s suggestions. “Growing evidence suggest that the microflora varies during the different tumoral stages and that microbes can influence disease progression. Lung cancer often progresses quickly. Sometimes lung cancer arises with multiple synchronous nodules or metastases can disseminate starting from a small parenchymal primary mass. It is thus relevant to evaluate the microbiome variation during cancer progression according to diagnostic, prognostic, and predictive perspectives. It has been also reported that a specific microbiome is detectable in early-stage lung cancer (identified at CT scan as ground-glass nodules GGN) carrying different taxa signatures when compared with that extracted from normal adjacent tissue . In addition, microbial dysbiosis is associated with tumor progression, clonal evolution, and spreading to distant sites . If malignant cells can model the microbiome , it has been also reported that distinct “dysbiotic signatures “ can act directly on tumor cells and indirectly through the suppression of immune response. Some experimental evidence documented the associations between Mycoplasma pn. infection and development of metastases in lung cancer. In immortalized human bronchial epithelial cells and A549 adenocarcinoma cell lines, Mycoplasma influences malignant transformation by activating the expression of the growth factor bone morphogenetic protein 2 (BMP2) “.
Comment 10. The paragraph between lines 343-378 contains a large amount of relevant and interesting information, but it contains no references except for one (ref. 115) at the end of the paragraph. Each sentence in the paragraph should be supported by relevant references.
Answer 10. We thank the Reviewer for this comment. The text has been extensively revised. Immune checkpoints identify molecules that are located on cell surface which act by sending inhibitory stimuli to reduce immune responses. The expression of these mole-cules is exploited by tumors to escape immune control as expedience to progress and dis-seminate . The most know checkpoints are the cytotoxic T-lymphocyte-associated an-tigen 4 (CTLA-4) and the programmed death 1 (PD-1) immune checkpoint and its ligand (PD-L1). The inhibitors of immune checkpoints (ICIs) are drugs that can block checkpoint proteins from binding with their ligands thus restoring immune response against cancer. The microbiome is implicated in response to immune checkpoint inhibitors (ICI) . In lung cancer mouse models it has been possible to show that Bifidobacterium bifidum strains synergistically cooperate with anti PD-L1 agents and oxaliplatin in reduc-ing tumor mass . In melanoma patients, fecal microbiota transplant (FMT) can re-store sensitivity to ICIs in previously anti PD-1 resistant tumors . Notably, re-sponse to ICI is associated with the activation of intestinal dendritic cells and circulating T cells by specific intestinal microbes. Moreover, some bacteria might increase sensitiv-ity to CTL-4 blockade based on the upregulation of Treg. Innate immunity is repre-sented by antigen-presenting cells (APCs), dendritic cells, and B lymphocytes. APC inter-acts with adaptive immunity and modulates the differentiation of T lymphocytes. In the airway epithelium, commensal bacteria modulate APCs to activate lymphocytes Th17 and to induce Th1 differentiation. The strong reciprocal interaction between the mi-crobiome and immune responses gives rise to unique bacterial signatures which reflected the outcome of patients treated with ICIs. In metastatic melanoma patients which re-spond to ICIs have high concentrations of Faecalibacterium Prausnitzii which resides in the gut and which ultimately modulates the immune system via the production of short-chain fatty acids (SCFAs). It has been also shown that for NSCLC, kidney and urothelial carcinoma response to ICI is associated with the abundance of Akkermansia Muciniphilia and Enterococcus Hirae, according to a quite cancer-specific manner. The gut microbi-ota seems to be able to activate commensal-specific memory T cells that in turn, cross-react with tumor-associated antigens. E. Hirae harbors a prophage encoding an MHC class I-binding protein that induces a CD8+ T cell response and cross-reacts with cancer anti-gens. This evidence could explain how E. Hirae might impact on ICI treatment. In conclusion, growing evidence suggests that cancer patients who are addressed to ICI treatment might be stratified based on their microbiome composition, which carries pre-dictive and prognostic potential. The identification of specific microbial signatures should therefore represent an additional therapeutic option to improve response to immunother-apy. In this perspective, several clinical trials are ongoing ( for details www.clinicaltrial.gov ). The observational NCT04107168 trial is aimed at evaluating the microbiome as a predictive marker in patients treated with different immune checkpoint inhibitors. Similarly, the observational NCT04711330 trial evaluates the role of the micro-biome in patients carrying locally advanced lung cancer and treated with durvalumab fol-lowing concurrent chemotherapy, in a clinical setting recalling that of the PACIFIC study. The microbiome is also involved in modulating responses to ICI-induced toxicities and as a marker that can be potentially exploited to improve quality of life . Dif-ferences in the gut microbiome between lung cancer patients and healthy controls have been reported to be related to response to ICI treatment as well as to the occurrence of ad-verse events and drug-related toxicities ( NCT03688347 trial) . Similarly the phase IV study NCT04636775 aims to determine in immunotherapy naïve lung cancer patients who are going to be treated with ICI if microbiome composition can be related either to the outcome and to toxic effects. The ongoing MicroDurva study ( NCT04680377) is a phase IV trial aimed at determining if the analysis of microbiome in lung cancer patients treated with durvalumab can predict patients’ response (toxicity). The prospective NCT04954885 trial has the goal to functionally analyse microbiome as predictive and prognostic mark-ers for lung cancer patients receiving combinatorial therapies, according to the INSIGNA protocol ( which compares pembrolizumab alone as a first-line treatment, followed by pemetrexed and carboplatin with or without pembrolizumab after disease progression).
Round 2
Reviewer 1 Report
The authors addressed all previous concerns
Reviewer 3 Report
The authors addressed all my concerns and the paper can be published in the present form.